# Thiamine dose response in human milk with supplementation among lactating women in Cambodia: study protocol for a double-blind, four-parallel arm randomised controlled trial

Kyly C Whitfield,[1] Hou Kroeun,[2] Tim Green,[3] Frank T Wieringa,[4] Mam Borath,[5] Prak Sophonneary,[6] Jeffrey R Measelle,[7] Dare Baldwin,[7] Lisa N Yelland,[3,8] Shalem Leemaqz,[3] Kathleen Chan,[1] Jelisa Gallant[1]

For numbered affiliations see end of article.

**Correspondence to**
Dr Kyly C Whitfield;
kyly.whitfield@msvu.ca

## ABSTRACT

**Introduction** Thiamine (vitamin B1) deficiency remains a concern in Cambodia where women with low thiamine intake produce thiamine-poor milk, putting their breastfed infants at risk of impaired cognitive development and potentially fatal infantile beriberi. Thiamine fortification of salt is a potentially low-cost, passive means of combating thiamine deficiency; however, both the dose of thiamine required to optimise milk thiamine concentrations as well as usual salt intake of lactating women are unknown.

**Methods and analysis** In this community-based randomised controlled trial, 320 lactating women from Kampong Thom, Cambodia will be randomised to one of four groups to consume one capsule daily containing 0, 1.2, 2.4 or 10 mg thiamine as thiamine hydrochloride, between 2 and 24 weeks postnatal. The primary objective is to estimate the dose where additional maternal intake of thiamine no longer meaningfully increases infant thiamine diphosphate concentrations 24 weeks postnatally. At 2, 12 and 24 weeks, we will collect sociodemographic, nutrition and health information, a battery of cognitive assessments, maternal (2 and 24 weeks) and infant (24 weeks only) venous blood samples (biomarkers: ThDP and transketolase activity) and human milk samples (also at 4 weeks; biomarker: milk thiamine concentrations). All participants and their families will consume study-provided salt *ad libitum* throughout the trial, and we will measure salt disappearance each fortnight. Repeat weighed salt intakes and urinary sodium concentrations will be measured among a subset of 100 participants. Parameters of $E_{max}$ dose–response curves will be estimated using non-linear least squares models with both 'intention to treat' and a secondary 'per-protocol' (capsule compliance ≥80%) analyses.

**Ethics and dissemination** Ethical approval was obtained in Cambodia (National Ethics Committee for Health Research 112/250NECHR), Canada (Mount Saint Vincent University Research Ethics Board 2017–141) and the USA (University of Oregon Institutional Review Board 07052018.008). Results will be shared with participants' communities, as well as relevant government

## Strengths and limitations of this study

► Human milk is thiamine-responsive; however, this is the first study to assess the dose of thiamine required to optimise human milk thiamine concentrations, vital information required to inform fortification programmes (minimise costs and unnecessary overages).

► Previous work has identified delayed cognitive development among children with low thiamine intake in early life; however, this will be the first study to explore cognitive development among infants with controlled thiamine exposures in the first 6 months postnatal.

► We will collect mother's blood, human milk and infant's blood at 24 weeks postnatal, enabling a robust evaluation of associations between maternal supplementation and these biomarkers.

► The study intervention begins at 2 weeks postnatal, so while randomisation should prevent pretreatment differences between treatment groups, we will not have biochemical thiamine status of the mother during pregnancy, nor the infant at birth.

► Household salt disappearance measures may overestimate or underestimate actual usual salt intake as it will not account for wastage, nor salt from food consumed outside the home (eg, with relatives, purchased at market), respectively.

and scientific stakeholders via presentations, academic manuscripts and consultations.
**Trial registration number** NCT03616288.

## BACKGROUND

Thiamine (vitamin B1) is an essential water-soluble micronutrient required for energy metabolism and nerve impulse conduction.[1 2] Thiamine deficiency was recently called the forgotten disease of Asia,[3]

as it remains an understudied public health issue despite evidence of suboptimal thiamine intake or status in the region,[4] including Cambodia.[5–11] Thiamine deficiency is of particular concern among lactating women because mothers with poor dietary thiamine intake and/or status produce milk low in thiamine, putting their exclusively breastfed infants at a high risk of developing infantile beriberi.[12] Infantile beriberi presents during the exclusive breastfeeding period and without treatment can result in death within hours of clinical presentation.[4 13] In addition, a growing body of evidence from an unfortunate 'natural experiment' in which thiamine was erroneously omitted from infant formula[14] suggests that thiamine deficiency not severe enough to cause clinical beriberi symptoms can negatively impact cognitive development and functioning.[15–17] To date, the precise pathways and mechanisms remain poorly understood, and important questions remain about the timing and levels of deficiency, as well as the ability of prophylactic or therapeutic interventions to prevent or remediate the effects of suboptimal thiamine status on cognitive outcomes in humans.[18]

Thiamine is found in pork, whole grains and legumes,[1] and must be consumed routinely due to the relatively short half-life and lack of body storage.[4] Lactating women have an estimated average requirement (EAR) and recommended dietary allowance (RDA) for thiamine of 1.2 and 1.4 mg/day, respectively.[19] For infants aged 0–6 months, an adequate intake (AI) of 0.2 mg/day was derived using the human milk thiamine concentrations from healthy, well-nourished mothers, assuming a daily milk intake of 780 mL.[19]

The dietary staple of Cambodia is B-vitamin poor, white, polished rice, which accounts for upwards of 60% of daily energy intake.[20] Given the challenges associated with supplementation and changing dietary patterns (recently reviewed in Whitfield *et al*[4]), fortification, a sustainable, cost-effective and passive intervention,[21–24] is a potentially suitable solution for improving maternal thiamine intake.[23] Thiamine has few technical constraints as a fortificant,[21] and there is no tolerable upper intake level (UL) for thiamine because there have been no reports of adverse effects of excess thiamine intake.[19 25 26] We recently demonstrated that maternal consumption of thiamine-fortified fish sauce significantly increased maternal, human milk, and infant thiamine status[8] However, centrally produced fish sauce may not reach the poorest communities who make their own fish sauce, and consumption of this condiment is not universal in all regions where we find thiamine deficiency.[27 28] Conversely, salt is a common condiment in most regions of the world, and has proven to be a successful global fortification vehicle for iodine.[29 30]

The overall aim of this study is to obtain the information necessary to formulate a thiamine-fortified salt for future use in Cambodia (ideal thiamine dose, usual salt intake), and to explore the impact of various doses of maternal thiamine on markers of infant cognitive development.

## STUDY OUTCOMES AND ANALYSIS OBJECTIVES

Primary outcome and analysis objective: To estimate the dose on the dose–response curve where additional maternal intake of thiamine (oral dose) no longer meaningfully increases human milk total thiamine concentration at 24 weeks postpartum.

### Secondary outcomes and analysis objectives

1. To estimate the dose on the dose–response curve where additional maternal intake of thiamine (oral dose) no longer meaningfully increases infant thiamine diphosphate concentrations (ThDP) 24 weeks postnatally, and assess whether this depends on the presence/absence of a genetic haemoglobin disorder.
2. To estimate the dose on the dose–response curve where additional maternal intake of thiamine (oral dose) no longer meaningfully increases human milk total thiamine concentration at 4 and 12 weeks postpartum.
3. To estimate the dose on the dose–response curve where additional maternal intake of thiamine (oral dose) no longer meaningfully increases infant transketolase activity at 24 weeks postnatally, and assess whether this depends on the presence/absence of a genetic haemoglobin disorder.
4. To estimate the dose on the dose–response curve where additional maternal intake of thiamine (oral dose) no longer meaningfully increases maternal ThDP at 24 weeks postpartum, and assess whether this depends on the presence/absence of a genetic haemoglobin disorder.
5. To estimate the dose on the dose–response curve where additional maternal intake of thiamine (oral dose) no longer meaningfully increases maternal transketolase activity at 24 weeks postpartum, and assess whether this depends on the presence/absence of a genetic haemoglobin disorder.
6. To test for differences between the four randomised groups on human milk total thiamine at 4, 12 and 24 weeks postpartum.
7. To test for differences between the four randomised groups on maternal ThDP at 24 weeks postpartum, and assess whether this depends on the presence/absence of genetic haemoglobin disorder.
8. To test for differences between the four randomised groups on maternal transketolase activity at 24 weeks postpartum and assess whether this depends on the presence/absence of a genetic haemoglobin disorder.
9. To estimate usual household salt intake from mean fortnightly salt disappearance (weight lost, in g).
10. To estimate salt intake among a subset of 100 lactating women, their male partners (if applicable), and their children 24–59 months (if applicable) using observed weighed salt intake records.
11. To estimate sodium intake using 24-hour urinary sodium concentrations among a subset of 100 lactating women.

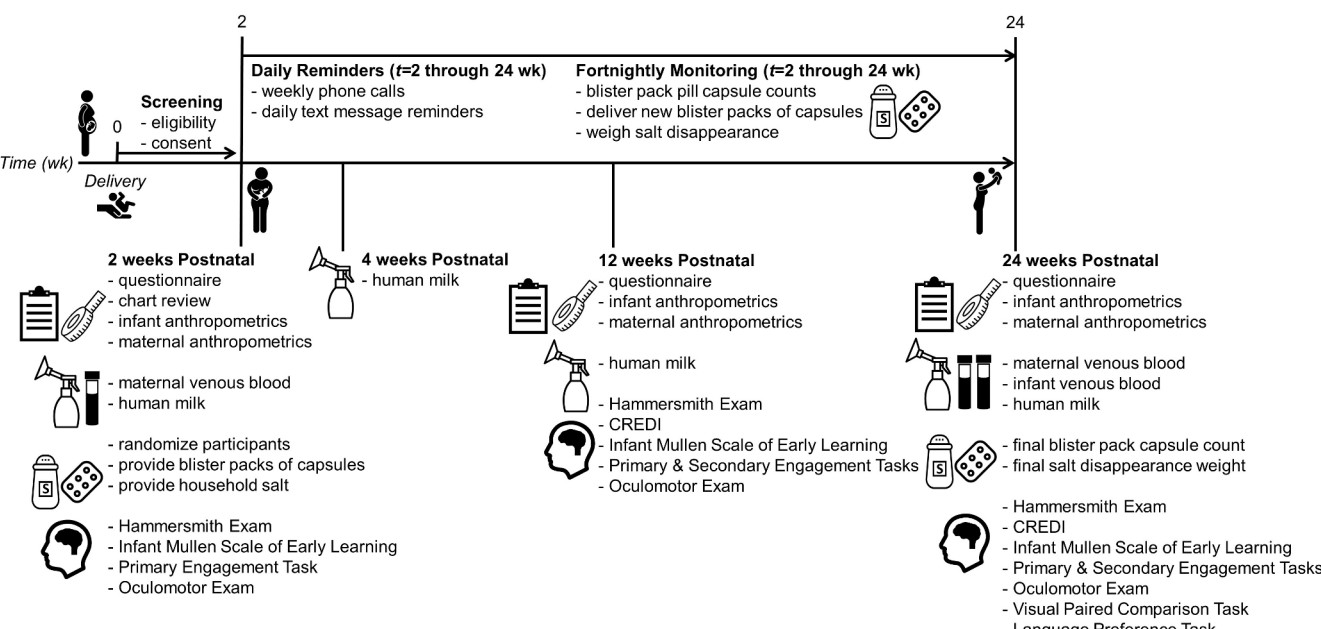

**Figure 1** Study timeline and data collection schedule for Trial of thiamine supplementation in Cambodia. CREDI, Caregiver Reported Early Development Instruments; wk, week.

12. To test for differences between infants in the 0 and 10 mg randomised groups on Composite Mullen and the five subscales of the Mullen at 24 weeks postnatally.

13. To test for differences between infants in the 0 and 10 mg randomised groups on Visual Paired Comparison Novelty Score and the attention and processing speed subscales at 24 weeks postnatally.

14. To test for differences between infants in the 0 and 10 mg randomised groups on the Language Preference Task Score at 24 weeks postnatally.

15. To test for differences between infants in the 0 and 10 mg randomised groups on oculomotor scores at 24 weeks postnatally.

16. To determine the effect of inflammation, as measured by C-reactive protein (CRP) and α−1-acid-glycoprotein (AGP) on maternal ThDP at 2 and 24 weeks postpartum, and infant ThDP at 24 weeks postnatal.

## METHODS
### Study design and setting
This is a double-blind, four-parallel arm randomised controlled trial among lactating women and their newborn infants. The study timeline and data collection schedule can be found in figure 1. The study is community-based, with data collection taking place in women's homes in Kampong Thom province in central Cambodia. All women live rurally, in villages in the catchment areas of the following health centres: Tboung Kapoeur, Kampong Svay, Sankor, Chey, Salavisai, Prey Kuy, Prey Pros and Srayov.

### Eligibility criteria
All women must provide written informed consent to participate. Participants must meet the following criteria:
► mothers of a newborn;
► aged 18–45 years;
► most recent pregnancy was normal (ie, no known chronic conditions, preeclampsia, gestational diabetes, etc), and the singleton infant was born without complications (eg, low birth weight (<2.5 kg), tongue tie, cleft palate);
► intends to exclusively breastfeed for 6 months;
► resides in Kampong Thom province, Cambodia, and is not planning to move in the next 6 months;
► is not currently taking, and has not taken any thiamine-containing supplements over the previous 4 months;
► is not currently participating in any nutrition programmes beyond normal care;
► is willing to consume one capsule daily from 2 weeks through 24 weeks postpartum;
► is willing for her entire household consume only salt provided by the study team; and
► is willing for the following biological samples to be collected: a maternal venous blood sample and human milk sample at 2 weeks postpartum, a human milk sample at 4 and 12 weeks postpartum and maternal and infant blood samples and a human milk sample at 24 weeks postpartum.

### Recruitment, allocation, randomisation and blinding
A participant flow diagram is shown in figure 2. Participants will be recruited through antenatal care visits, as well as through consultation with local village chiefs, elders and health centre staff. Pregnant women will be

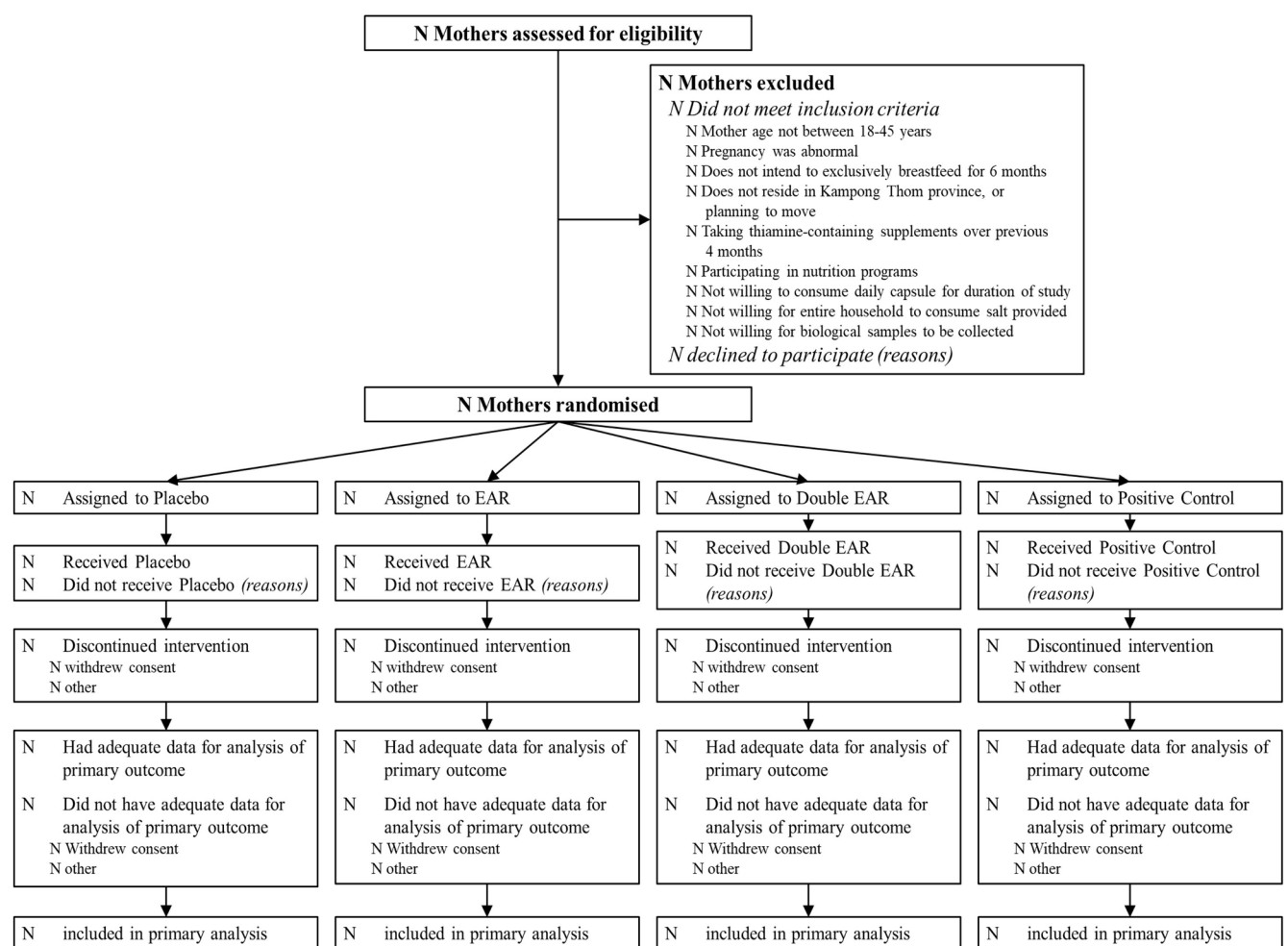

**Figure 2** Participant flow chart for *Trial of thiamine supplementation in Cambodia.* Double EAR, Double EAR group (2.4 mg thiamine as thiamine hydrochloride); EAR, EAR group (1.2 mg thiamine as thiamine hydrochloride); Placebo, negative control group (placebo; 0 mg thiamine); Positive Control, Positive Control group (10 mg thiamine as thiamine hydrochloride).

advised and provided with a general overview of the research study, and then the names and contact information of interested women will be shared with the research assistants, who follow up after delivery to screen for eligibility, obtain consent and enrol women. Recruitment will continue on a rolling basis until 320 women are enrolled.

Women will be randomly assigned to one of the four treatment groups as described in table 1. Participants, research assistants, study investigators and data analysts will be blinded to the randomised groups. The randomisation will be stratified by health centre and use randomly permuted blocks of size 8 within strata to assign

participants to one of eight treatment codes in the ratio 1:1:1:1:1:1:1:1 (two treatment codes per treatment group to assist with blinding). A computer-generated randomisation schedule was prepared by the study statisticians using ralloc.ado in Stata (College Station, Texas, USA). An independent scientist determined which treatment code corresponded to which treatment group. When a participant is enrolled in the study, the research assistants open an envelope labelled with the study ID that contains the study ID and accompanying treatment code that will be assigned to that participant.

| Table 1 | Treatment arms for the *Trial of thiamine supplementation in Cambodia* | |
| --- | --- | --- |
| **Treatment arm** | **Thiamine dose** | **Rationale** |
| Negative control | 0 mg/day | Negative control (placebo) |
| EAR | 1.2 mg/day | 1×thiamine EAR for lactating women[19] |
| Double EAR | 2.4 mg/day | 2×thiamine EAR for lactating women[19] |
| Positive control | 10 mg/day | Positive control (dose currently given in supplemental form in Myanmar)[4] |

EAR, estimated average requirement.

## Intervention

Women will be asked to consume one capsule daily between 2 and 24 weeks postnatal. The intervention is opaque capsules containing varying amounts of thiamine hydrochloride and cellulose filler, as indicated in table 1. All thiamine is delivered as thiamine hydrochloride, calculated using a 1.271 correction factor (ratio of molecular weights of thiamine hydrochloride and thiamine). Capsules were formulated, compounded and packaged as 14 day blister packs at the Quinpool Wellness Centre in Halifax, Nova Scotia, Canada.

All capsules were formulated and packed in two batches in July and November 2018. Thiamine content was assessed by an independent laboratory (USANA, Salt Lake City, Utah, USA) before distribution to participants, and will continue to be assessed bimonthly for the duration of the study. Capsules will be deemed acceptable if the average thiamine concentration for each code (n=10 samples per code) falls within ±15% of the values shown in table 1.

## Compliance

Compliance will be assessed fortnightly: research assistants will visit the participant's home to collect the old blister pack and complete a capsule count, and to deliver a new blister pack. A woman will be considered compliant if she consumes ≥80% capsules over the 22-week intervention.

## Participant and public involvement in research

This research is being conducted in collaboration with the Cambodian Ministry of Health's National Maternal and Child Nutrition Programme, the Cambodian Ministry of Planning's Sub-Committee for Food Fortification and Kampong Thom province's Provincial Health Department and Health Operational District. Participants and the public will be involved in the dissemination of study results (public village meetings to share findings), however were not involved in research question development, nor the design of the study or intervention.

## DATA AND BIOLOGICAL SAMPLE COLLECTION

Data and biological samples will be collected at enrolment (between $t$=0 to 2 weeks postpartum), at 2, 4, 12 and 24 weeks postnatal as well as during fortnightly monitoring visits (see figure 1).

## Chart review

Upon enrolment, a healthcare worker in the Health Centre will record data from the dyad's chart such as any interventions/events at birth, maternal age, number of antenatal care visits, number of iron-folic acid tablets consumed, time and date of birth, sex of infant, and infant anthropometrics.

## Questionnaires

Using an interviewer-administered questionnaire, we will collect demographic and socioeconomic information, health information including questions about sleep (adapted from[31]) and postpartum depression,[32] limited dietary intake data (eg, perceptions of salt intake, postpartum food taboos[33]) and infant and young child feeding (IYCF) knowledge and behaviours at 2, 12 and 24 weeks postnatal.

## Anthropometry

Anthropometric measures are as follows: infants: length, weight and head circumference; mothers: height and weight. Initial measurements will be taken in the Health Centre at delivery, and all other measurements will be collected in participant's homes (2, 12 and 24 weeks postnatal) using calibrated instruments and standard protocols as per.[34]

## Fortnightly monitoring

Every 2 weeks, research assistants will visit the participant's home to distribute a new blister pack and assess intervention compliance. They will also administer a short questionnaire about selling or sharing of salt, IYCF practices and the household members eating from the common household pot.

## Salt disappearance

At the initial home visit after randomisation, research assistants will check that the household has removed all salt from the home, and will distribute table salt in specialised study containers. Using calibrated scales (1 g graduation), research assistants will log the initial weight of the salt container(s). Participants will be asked to consume only salt provided, and since Cambodian families eat from a common pot, they will be instructed for all family meals to be prepared using this salt. Salt disappearance will continue to be assessed at fortnightly visits.

We will also collect supplemental information in a subset of households (n=100) using observed weighed intakes and maternal urinary sodium concentrations (randomly selected from larger study, with a second day repeat on non-consecutive days within 1 week of initial visit). The participant (mother), the husband/man in household aged 18–50 years and a child between 24–59 months (husband/man and child will not be applicable in all households) will be enrolled in this substudy. Research assistants will sit in the participants home from dawn through dusk, recording, at the individual level, intake for the 1–3 individuals for all table salt and salt-containing condiments (eg, fish and soy sauces) consumed. For the mothers only, we will assess 24-hour urinary sodium concentrations.[25] The mother will discard her first urine on awakening, record the time and then collect all urine over the next 24 hours into a provided container; the collection will end after the woman's first void the following morning. The research assistants will visit her home to weigh the full container and then take aliquots for testing. The urine sample will be temporarily stored at −20°C in Kampong Thom for <2 weeks before being transported to the National Institute for Public Health Laboratory (NIPHL) in Phnom Penh for urinary sodium

analysis on a EasyLtye Na/K/Cl Analyser (Medica, Dusseldorf, Germany) and urinary creatinine assessment using a Kenza 240TX Analyser (Biolabo Diagnostics, Maizy, France). We will model thiamine fortification of salt using the University of Iowa's Intake Monitoring, Assessment and Planning Program software (http://www.side.stat.iastate.edu/imapp.php).

### Venous blood samples

Trained, Khmer-speaking nurses will meet mothers and infants at their home or a central village location (health centre, or village chief's home) to collect maternal and infant blood samples into EDTA-coated tubes. Maternal blood samples (9 mL) will be collected at 2 and 24 weeks postnatal; infant samples (5 mL) will be collected only at 24 weeks postnatal. Time of day and time since last meal will be recorded.

### Human milk samples

Human milk samples will be collected using a battery-powered single breast pump (Swing Breast pump, Medela) at 2, 4, 12, and 24 weeks postnatal. One full breast expression (single breast) will be collected from the breast women self-identify as being more 'full' (the breast not most recently emptied). Since time of day has little effect on milk thiamine concentrations,[35] samples can be collected at any time of day, however, time of day, time since last meal, and breast side will be recorded.

### Blood and human milk processing

Biological samples will be collected in the village, placed on ice, and transported to the field lab in Kampong Thom within 5 hours of collection for processing. All samples will be stored at −20°C for <2 weeks before being transported to the NIPHL for storage at −80°C. Samples will be batch-shipped on dry ice after the 24 weeks postnatal data collection is completed.

### Blood and human milk analysis

Venous blood samples will undergo analysis for both transketolase activity and thiamine diphosphate concentrations (ThDP) at the NIHR BRC Nutritional Biomarker Laboratory at the University of Cambridge in the United Kingdom. At 2 weeks postnatal, maternal whole blood ThDP and erythrocyte transketolase activity will be measured. At 24 weeks postnatal, maternal ThDP and transketolase activity will be measured in both whole blood and erythrocytes, and infant samples will be assayed for whole blood ThDP and erythrocyte transketolase activity. When ThDP is assessed in whole blood, concentrations must be normalised to haemoglobin concentrations and/or hematocrit[36] measured via a HemoCue 201 portable hemoglobinometer and capillary hematocrit tubes, respectively.

Evidence from the 2014 Cambodian National Micronutrient survey indicates that eThDP is affected by the presence/absence of genetic haemoglobin disorders (unpublished data). With this, it is vital to assess genetic haemoglobin disorders, as this could influence

participant's response to thiamine, it may influence transketolase, and it could change thiamine dosage requirements in other countries. Therefore, at one timepoint only (2 weeks postnatal for maternal, and 24 weeks postnatal for infant samples), samples will undergo haemoglobin capillary electrophoresis analysis to identify structural haemoglobin variants.[37] Buffy coat samples will be stored in the BioBank in the Department of Applied Human Nutrition at Mount Saint Vincent University in Halifax, Nova Scotia, Canada, for potential later assessment of genetic haemoglobin disorders.

Biomarkers may be influenced by inflammation, which is common in this population.[38 39] Plasma samples will be sent to Dr. Jurgen Erhardt at the VitMin Lab in Germany for analysis of C-reactive protein (CRP) and α−1-acid-glycoprotein (AGP) using an immunosorbent assay.[40] Note that this assay will also measure retinol binding protein (RBP), ferritin, and soluble transferrin receptor (sTfR). Other plasma samples will be stored in the BioBank for potential future use.

The weight of the full human milk expression will be recorded, the sample mixed, and 2 mL aliquots obtained in amber cryovials. Human milk thiamine concentrations will be measured at the USDA/ARS Western Human Nutrition Research Center, University of California, Davis.[41] Additional human milk samples will be stored in the BioBank for potential future use.

### Cognitive assessments

We will conduct cognitive assessments at 2, 12, and 24 weeks postnatal; see figure 1 for battery of assessments at each timepoint. Assessments include: the Hammersmith Infant Neurological Exam, Infant Mullen Scales of Early Learning, a Primary Engagement Task, a Secondary Engagement Task, the Caregiver Reported Early Development Instruments (CREDI), an oculomotor exam, the Visual Paired Comparison task, and the Language Preference task.

The Hammersmith Infant Neurological Exam is a standard tool for providing basic information about infants' neurological status through use of gentle touch and gentle social interaction in order to examine infants' sensory and motor responses; it has previously shown excellent interobserver reliability even among less experienced staff.[42] The Infant Mullen Scales of Early Learning is an individually administered, multi-domain measure of early development, with scales measuring development in visual reception, fine motor, gross motor, receptive language, and expressive language.[43] The Primary Engagement Task measures individual variation in infant, caregiver, and dyad-level developmental change in the ability to engage contingently with the partner in direct positive mutual engagement interactions. The Secondary Engagement Task measures individual variation in caregiver and infant ability to engage mutually in relation to an external object. The CREDI is a caregiver-reported measure of child development that has been shown to have right validity and reliability in low-resource settings.[44] An oculomotor exam has

been included because oculomotor disturbances are not uncommon among thiamine deficient adults presenting with Wernicke's encephalopathy.[45] The Visual Paired Comparison task probes infants' recognition memory and attention, and has been used to predict subsequent verbal IQ in middle childhood,[46 47] and produce measures of attention predictive of developmental maturity and better cognitive function.[48 49] The Language Preference Task is designed to measure individual variation in infants' level of interest in the kind of language that caregivers typically direct toward infants (aka infant-directed talk or 'motherese') relative to a) the kind of language that is typically directed toward adults (aka adult-directed talk), and b) sounds that are matched in complexity but are non-linguistic (non-linguistic analogue). Over a series of trials, infants hear these three different kinds of sound samples through an audio speaker (infant-directed talk, adult-directed talk, non-linguistic analogue), and the duration that they look toward a visual display during each sound sample is subsequently measured from the videotaped record collected during the session.

### Participant remuneration

All participants will receive a mobile phone and mobile phone credits. In addition, modest, study-appropriate remuneration such as a sarong or laundry soap will be provided at biological sample collection points (2, 4, 12, and 24 weeks postnatal, and after urine collection).

### Safety considerations, safety monitoring, and breaking the blinding

There is no UL for thiamine,[19 25 26] and therefore no necessity for a Serious Adverse Events Committee or a Data Safety Monitoring Board in the current study. While we are confident that there is a very low risk of serious adverse events in this study related to the intervention, we do expect some infant deaths in our cohort due to other causes. Per the 2014 Cambodian Demographic and Health Survey rates of neonatal (birth to 1 month) and infant (birth to 1 year) mortality,[50] we calculated an expected mortality rate in our study to lie between 1.67 and 3.68 infant deaths.

The master ID list will not be unblinded during the study unless in the unlikely event of an adverse event in the trial. If there is a medical emergency and unblinding of a participant is required, our blinding mechanism would allow for only one of the two codes per treatment arm to be revealed (as each treatment arm will have two codes). We would then document and report to the funding agency and all ethics committees why any premature unblinding occurred.

Once data collection and cleaning is complete, the database will be locked and unblinded treatment codes will be included in the database because a blinded analysis is not possible for estimating dose response curves.

### Sample size calculations

To detect a clinically meaningful difference of 40 µg/L in human milk total thiamine concentration between any two treatment groups with 90% power, assuming an SD of 43 µg/L (estimated SD of control group in a Cambodian thiamine-fortified fish sauce trial[51]), 48 women are required per treatment group, or a total of 192 women. This sample size allows for 20% attrition and uses a two-sided alpha of 0.0083 for each of the six pairwise comparisons between the four treatment groups in order to control the familywise error rate at the 0.05 level using a Bonferroni adjustment for multiple comparisons. Recruitment of 320 participants (80 per group) was planned to allow for some uncertainty in the assumed values, particularly the SD which may be larger than anticipated.

Simulations of 500 dose-response curves for human milk total thiamine concentration were conducted to estimate the precision that this sample size would provide for addressing the primary study objective (ie, for estimating the dose on the dose response curve where additional maternal intake of thiamine (oral dose) no longer meaningfully increases human milk total thiamine concentration at 24 weeks postpartum, defined as the dose required to achieve 90% of the average maximum human milk total thiamine concentration). Data were simulated based on an average minimum and maximum concentration of 136 µg/L[51] and 210 µg/L,[19] respectively, with an SD of 43 µg/L,[51] and reaching 50% and 90% of the maximum average concentration at dose 1.2 mg/d (EAR group) and 2.4 mg/d (double EAR group), respectively. Assuming an $E_{max}$ dose–response curve, the precision of the estimated dose is ±1.29 mg/d (ie, the 95% CI for the estimated dose will be within ±1.29 mg/d of the point estimate).

### Data and sample management and security

Participants will be given a unique alpha-numeric study ID code, not derived from personal identifiers; this code will link all data collected from this individual. All data will be collected directly on tablets using structured forms designed specifically for this study. Data will be reviewed daily by the research assistant and field supervisor before secure download and integration with the larger database. Data management and security procedures, including assurance of confidentiality, adhere to the Canadian Tri-Council Policy Statement on Ethical Conduct for Research Involving Humans (TCPS2 CORE) guidelines, and are outlined in full in the protocol at clinicaltrials.gov (NCT03616288).

### Data analysis plan

A full statistical analysis plan is published in the clinicaltrials.gov entry (NCT03616288). The primary analyses will be performed on an 'intention to treat' basis, where all participants will be analysed according to their allocated treatment group regardless of compliance. A secondary 'per-protocol' analysis will be performed on the subset of women who consumed ≥80% capsules over the study period. The parameters of the $E_{max}$ dose–response curves will be estimated using non-linear least squares models. Separate curves will be fitted for outcomes measured at 4 weeks, 12 weeks and 24 weeks postnatal. The dose where

additional maternal intake of thiamine no longer meaningfully increases milk thiamine concentration, defined as the dose that achieves 90% of the maximum concentration, will be estimated from the fitted curve with SE estimated by bootstrapping for calculating 95% CI.

Comparisons between treatment groups will be performed using linear mixed-effects models to account for repeated measurements and adjusted for randomisation strata and levels at 2 weeks postnatal. Interaction tests will be performed to assess whether treatment effects depend on genetic haemoglobin disorder. Missing data will be addressed using multiple imputation to create 100 complete data sets for analysis, with a sensitivity analysis performed on the raw (unimputed) data.[52]

### Ethics and dissemination
Approvals were obtained. This study is registered at clinicaltrials.gov (ClinicalTrials.gov Identifier NCT03616288 (7 August 2018); Protocol version 2.1 (24 July 2018)).

Updates on the study will be shared at the monthly Cambodian Nutrition Working Group meetings, attended by researchers and non-governmental organisations (NGO) engaging in nutrition research and programming in Cambodia. Scaling Up Nutrition and other relevant groups will also be updated regularly. Research results will be presented at academic nutrition, public health and psychology conferences, and in peer-reviewed journals that offer open access. Data will be made available on a public repository after dissemination.

We plan to create various lay outputs from this study that can be used by NGO and government agencies working with families, or can be accessed directly by families. We will host a Dissemination Workshop in Phnom Penh open to relevant stakeholders (NGOs, government, researchers, media, clinicians, public health, all sectors) to share the main outcomes of the study, and officially 'launch' lay resources. We will also return to each community in Kampong Thom to conduct a village-wide meeting relaying study results.

## TRIAL STATUS
Participant recruitment began in August 2018, and data collection started on 12 September 2018. We estimate data collection will be completed by June 2019.

## DISCUSSION
Suboptimal maternal thiamine intake and/or status puts exclusively breastfed infants at risk of low thiamine status,[12] impaired cognitive development[15–17] and infantile beriberi, which can be fatal.[13] Thiamine fortification of salt is a potentially low-cost and sustainable means of combating suboptimal thiamine status; however, knowledge gaps must be filled before thiamine fortification can proceed.[4] There are limited data available on the dose of thiamine required by lactating women to optimise the thiamine concentrations in their milk. Further, usual salt intake among lactating women is unknown. Finally, although there is emerging evidence that low thiamine intake in early life impacts cognitive development, this has yet to be assessed in a controlled research environment. This study is posed to address these knowledge gaps and provide valuable information to inform any potential future thiamine fortification efforts in Cambodia or bordering countries.

**Author affiliations**
[1]Mount Saint Vincent University, Halifax, Nova Scotia, Canada
[2]Helen Keller International Cambodia, Phnom Penh, Cambodia
[3]South Australian Health & Medical Research Institute, Adelaide, Australia
[4]UMR-204 Nutripass, Institut de Recherche pour le Développement, UM/IRD/Supagro, Montpellier, France
[5]National Sub-Committee for Food Fortification, Cambodia Ministry of Planning, Phnom Penh, Cambodia
[6]National Nutrition Programme, Maternal and Child Health Centre, Cambodia Ministry of Health, Phnom Penh, Cambodia
[7]Department of Psychology, University of Oregon, Eugene, Oregon, USA
[8]School of Public Health, University of Adelaide, Adelaide, South Australia, Australia

**Acknowledgements** We thank the members of the study's Scientific Advisory Board for valuable input on the study protocol (alphabetical order): Drs. Megan Bourassa, Levente Diosady, Lisa Houghton, Arnaud Laillou and Annie Wesley. We thank Dr. Ken Brown for assistance and insight during study conception, Dr. Geraldine Richmond for her role in early protocol development and Keith Porter for early input on project feasibility and implementation. We acknowledge USANA (Salt Lake City, Utah, USA) for the generous provision of laboratory analysis for study capsule thiamine hydrochloride content.

**Contributors** KCW drafted the manuscript. KCW, HK, TG, FTW, JRM and DB conceived the study and wrote the initial study protocol. MB, SP, LNY, SL, KC and JG assisted in developing the protocol. HK, MB & SP facilitated implementation of the study. LNY and SL developed the statistical analysis plan. KC and JG are involved in study coordination. All authors participated in, read and approved the final manuscript.

**Funding** This study is funded through the Bill & Melinda Gates Foundation and the Sackler Institute for Nutrition Science, New York Academy of Sciences (Opportunity ID OPP1176128) as 'Objective 3: Trial of thiamine supplementation in Cambodia', as part of the larger program grant entitled 'Improving estimates of the global burden of thiamine deficiency disorders (TDDs) and approaches to their control'. The study funders were involved in early study conception. LNY was supported by an Australian National Health and Medical Research Council Early Career Fellowship (ID 1052388).

**Disclaimer** Funders will have no involvement in the study design, collection, management, analysis, and interpretation of data, writing of the report, nor the decision to submit the report for publication.

**Competing interests** None declared.

**Patient consent for publication** Not required.

**Ethics approval** National Ethics Committee for Health Research, Cambodia (112/250NECHR), Mount Saint Vincent University Research Ethics Board, Canada (2017 – 141); and the University of Oregon Institutional Review Board, USA (07052018.008)

**Provenance and peer review** Not commissioned; externally peer reviewed.

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
