## [Reviewer comments · BMJ Open]

ARTICLE DETAILS

TITLE (PROVISIONAL)	Thiamine dose response in human milk with supplementation among lactating women in Cambodia: study protocol for a double-blind, four-parallel arm randomized controlled trial
AUTHORS	Whitfield, Kyla; Kroeun, Hou; Green, Tim; Wieringa, Frank; Borath, Mam; Sophonneary, Prak; Measelle, Jeff; Baldwin, Dare; Yelland, Lisa; Leemaqz, Shalem; Chan, Kathleen; Gallant, Jelisa

VERSION 1 - REVIEW

REVIEWER	Tafere G Belay Central Washington University, Ellensburg, WA USA
REVIEW RETURNED	30-Jan-2019

GENERAL COMMENTS	The title was clearly written on the abstract but not in the main document. The title in the main document doesn't include the design. Participant timeline and informed consent material are not attached
---

REVIEWER	Maria Lorella Gianni Associate Professor, Department of Clinical Science and Community Health, University Study of Milan, Italy
REVIEW RETURNED	18-Mar-2019

GENERAL COMMENTS	I congratulate the authors for this interesting study protocol. It will be interesting to see the related results published
---

REVIEWER	Daniela USDA/ARS Western Human Nutrition Research Center and UC Davis USA
REVIEW RETURNED	01-Apr-2019

GENERAL COMMENTS	The manuscript under review by Whitfield et al. is well written and should be considered for publication after some minor comments have been addressed:
---

	 - L43ff: if the word count allows it would be very informative to add the actual time points for sample collection of the specimen. - L165-176: While the test and the time of administering the test has been noted, please also add the target group (infant, mothers, etc...) for clarification for the reader who is less familiar with these approaches. - L180ff: Please add your study population to the Study design and setting paragraph. - Eligibility: are the women apparently healthy? Any restrictions on illness? Chronic diseases? - L243ff: any questions about who consumed the missing capsules? They could be shared as well.
--	--

VERSION 1 – AUTHOR RESPONSE

Reviewer: 1

The title was clearly written on the abstract but not in the main document. The title in the main document doesn't include the design.

The title of the manuscript submitted is: "Thiamine dose response in human milk with supplementation among lactating women in Cambodia: study protocol for a double-blind, four-parallel arm randomized controlled trial".

The study design is re-iterated in the main text body under 'Methods' □ 'Study design and setting' □ "This is a double-blind, four-parallel arm randomized controlled trial..."

Participant timeline and informed consent material are not attached

The participant timeline is included as Figure 1. We have attached the study consent form as Supplementary File for this re-submission.

Reviewer: 2

I congratulate the authors for this interesting study protocol. It will be interesting to see the related results published.

Thank you, and thank you for reviewing the protocol manuscript.

Reviewer: 3

The manuscript under review by Whitfield et al. is well written and should be considered for publication after some minor comments have been addressed:

- L43ff: if the word count allows it would be very informative to add the actual time points for sample collection of the specimen.

The abstract word count is 300, and the original submission had exactly 300 words. We have added the timepoints in red-coloured text, however, since different data/samples are collected at different timepoints, this brought the abstract to 317 words.

- L165-176: While the test and the time of administering the test has been noted, please also add the target group (infant, mothers, etc...) for clarification for the reader who is less familiar with these approaches.

The secondary outcomes listed here are all cognitive development and neurological tests that will administered with the infants in the study. We have updated the outcomes accordingly; please see red-coloured text.

- L180ff: Please add your study population to the Study design and setting paragraph.

Thank you. We have updated the first sentence of this paragraph to read: "This is a double-blind, four-parallel arm randomized controlled trial among lactating women and their newborn infants."

- Eligibility: are the women apparently healthy? Any restrictions on illness? Chronic diseases?

The eligibility criteria are as currently outlined in lines 190-206: the woman's pregnancy had to be 'normal', without any complications or illnesses (e.g. preeclampsia, gestational diabetes). For your interest, we did not specify that women had to be healthy/free of chronic conditions; the reasoning for this is two-fold: 1) that chronic conditions are not always diagnosed in rural areas due to a lack of access to healthcare, and 2) because we wanted to conduct a pragmatic trial encompassing lactating women in rural Cambodia.

- L243ff: any questions about who consumed the missing capsules? They could be shared as well.

When the research assistants visit the participant's home every two weeks, they are tasked with collecting the old blister pack and completing a short fortnightly questionnaire to record data on compliance (as well as salt disappearance). This questionnaire captures data such as taking multiple capsules in one day, reasons for not taking capsules that remain, and has 'other' options to capture instances such as sharing capsules. Small details such as these have not been included in the manuscript due to word count restraints.